# HLGNN-MDA: Heuristic Learning Based on Graph Neural Networks for miRNA–Disease Association Prediction

**DOI:** 10.3390/ijms232113155

**Published:** 2022-10-29

**Authors:** Liang Yu, Bingyi Ju, Shujie Ren

**Affiliations:** School of Computer Science and Technology, Xidian University, Xi’an 710071, China

**Keywords:** miRNA–disease association, graph neural network, graph classification, heuristics learning

## Abstract

Identifying disease-related miRNAs can improve the understanding of complex diseases. However, experimentally finding the association between miRNAs and diseases is expensive in terms of time and resources. The computational screening of reliable miRNA–disease associations has thus become a necessary tool to guide biological experiments. “Similar miRNAs will be associated with the same disease” is the assumption on which most current miRNA–disease association prediction methods rely; however, biased prior knowledge, and incomplete and inaccurate miRNA similarity data and disease similarity data limit the performance of the model. Here, we propose heuristic learning based on graph neural networks to predict microRNA–disease associations (HLGNN-MDA). We learn the local graph topology features of the predicted miRNA–disease node pairs using graph neural networks. In particular, our improvements to the graph convolution layer of the graph neural network enable it to learn information among homogeneous nodes and among heterogeneous nodes. We illustrate the performance of HLGNN-MDA by performing tenfold cross-validation against excellent baseline models. The results show that we have promising performance in multiple metrics. We also focus on the role of the improvements to the graph convolution layer in the model. The case studies are supported by evidence on breast cancer, hepatocellular carcinoma and renal cell carcinoma. Given the above, the experiments demonstrate that HLGNN-MDA can serve as a reliable method to identify novel miRNA–disease associations.

## 1. Introduction

Since the discovery of microRNAs, an increasing number of researchers have investigated these molecules [1,2,3,4,5,6,7,8]. In particular, the discovery of a regulatory role for microRNAs in cellular activity suggests that these molecules are inextricably linked to many diseases [9,10,11,12,13]. Uncovering microRNA–disease associations has important implications for understanding disease mechanisms and assisting disease treatment [14,15,16,17,18,19,20]. However, due to the long time period required for biological experiments and the high resource costs, the use of computational methods to predict miRNA–disease associations has now become an important means for guiding traditional biological experiments, and it has greatly improved the efficiency of discovering disease-related miRNAs.

Some miRNA–disease association prediction models derive from combinatorial optimization theory and metric learning ideas, such as matrix-related operations and score estimation. MCMDA (matrix completion for miRNA–disease association) [21] performs matrix completion by applying a singular value thresholding algorithm on known miRNA–disease associations, and ILRMR (improved low-rank matrix recovery) [22] improves low-rank matrix recovery by referencing a weight matrix to enhance the prediction accuracy. MDMF (miRNA–disease Based on Matrix Factorization) [23] uses matrix factorization with disease similarity constraints to identify potential miRNA–disease associations. MDHGI (Decomposition and Heterogeneous Graph Inference) [24] discovers new miRNA–diseases associations by integrating the predicted association probability obtained from matrix decomposition through the sparse learning method. IMIPMF (inferring miRNA–disease interactions using probabilistic matrix factorization) [25] is a novel method for predicting miRNA–disease associations using probabilistic matrix factorization. WBSMDA (Within and Between Score for MiRNA–Disease Association prediction) [26] calculates a “Within and Between Score” for each miRNA–disease pair to predict the association between them. MLMD (Metric Learning for predicting miRNA–Disease) [27] is a novel computational model of metric learning for predicting miRNA–disease associations. It aims at learning miRNA–disease metrics to unravel not only novel disease-related miRNAs but also miRNA–miRNA and disease–disease similarities. DBNMDA (deep-belief network for predicting miRNA–disease associations) [28] constructs feature vectors to pre-train restricted Boltzmann machines for all miRNA–disease pairs and applies positive samples and the same number of selected negative samples to fine-tune a deep-belief network to obtain the final predicted scores.

Machine learning is also a class of methods [29,30] that are commonly applied to predict miRNA–disease associations [20,31,32,33,34,35,36,37]. RBMMMDA (restricted Boltzmann machine for multiple types of miRNA–disease associations) [38] proposes the restricted Boltzmann machine model to predict various types of miRNA–disease associations.

With the development of graph neural networks and the accumulation of large-scale graph data, in addition to traditional machine learning algorithms, DGCNN (multi-view multi-layer convolutional neural network) [39] and other deep learning [40,41,42,43,44,45] models have also been developed to deal with similar tasks. DGCNN focuses on large-scale and irregular network structures and adapts to the dynamic structure of local regions in the graph by flexibly designing convolutional filters. DeepMDA (predict miRNA–disease associations using deep learning) [46] uses a stacked self-encoder to obtain low-dimensional features from two high-dimensional feature vectors of miRNAs and diseases. A three-layer deep neural network [47] has then been developed to train classifiers of miRNA–disease feature pairs. MDA-CNN [48] constructs a three-layer miRNA–gene–disease association (MDA) network, and the network-based features of miRNAs and diseases are extracted using genes as the intermediate medium. The features are then downscaled using a self-encoder. Convolutional neural networks (CNNs) [49] are then used to further learn features from the miRNA–disease feature pairs. MDA-GCNFTG [50] predicts associations based on graph convolutional networks via graph sampling through the feature and topology graph to improve the training efficiency and accuracy. Instead of using heterogeneous graphs, MDA-GCNFTG constructs a homogeneous graph with MDPs (miRNA–disease pairs) as the nodes, which is the biggest difference with respect to our method. Although both use the GCN algorithm, the background graphs and the model focused on are completely different. For this task, the models focus on solving different problems.

However, there are still shortcomings in these recently proposed excellent computational models. Most of the current methods for predicting miRNA–disease associations are based on a strong assumption of similarity data. However, different models have different definitions of similarity, which makes the prediction results inaccurate. In addition, the miRNA functional similarity is incomplete and derived from known associations. Additionally, inconsistencies and incompleteness further lead to inaccurate prediction results. In this article, a heuristic learning method based on graph neural networks for miRNA–disease association prediction (HLGNN-MDA) is proposed. Inputting the whole miRNA–disease association network into the graph neural network for the model training causes high computational costs. To overcome this, we choose to train the graph neural network on enclosing subgraphs.

Figure 1 shows the overall framework of HLGNN-MDA. Our HLGNN-MDA model improves the graph neural network so that it can learn the information between miRNA and disease nodes and the topological relationships among homogeneous nodes at the same time. Compared with the previous computational models, our proposed method does not require too many similarity data and improves the applicability of graph neural networks in bipartite graph networks. More importantly, under such conditions, HLGNN-MDA can also achieve more accurate predictions.

**Figure 1 ijms-23-13155-f001:**
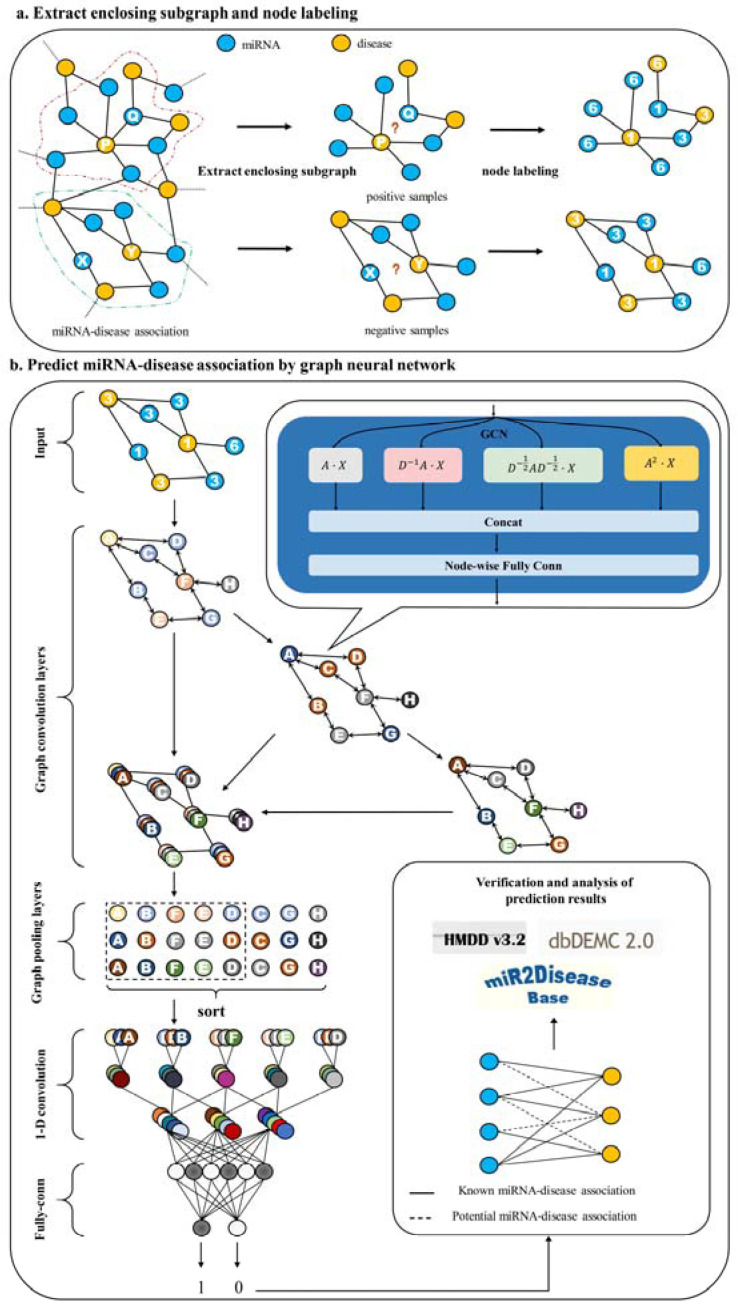
Flowchart of HLGNN-MDA. (**a**) The enclosing subgraph of all pairs of nodes are extracted, and all the nodes in each enclosing subgraph are labeled. (**b**) The enclosing subgraphs are input into the graph neural network. The graph convolution layers adopt the three-layer structure model as shown in Figure 2 where each sub-figure of three in the second part corresponds to the processing results of three convolution modules. As shown in Figure 3, each convolution module has the same structure. The prediction results are obtained through the graph convolution layer, the graph pooling layer, 1D convolution and fully connected layers. Finally, the predicted results are verified against databases.

## 2. Results and Discussion

### 2.1. Performance Analysis of HLGNN-MDA Mode

In this section, we compare HLGNN-MDA with other related methods using tenfold cross-validation [20]. The algorithms selected for comparison were BLHARMDA (bipartite local models and hubness-aware regression for miRNA–disease association prediction) [51], BNPMDA (bipartite network projection for miRNA–disease association prediction) [52], IMCMDA (inductive matrix completion for miRNA–disease association prediction) [52], LFEMDA (predict miRNA–disease associations by latent features extraction) [53] and MKRMDA (multiple kernel learning-based Kronecker regularized least squares for miRNA–disease association prediction) [54]. All models for performance analysis experiments adopted the association data from HMDD v2.0. We also performed a performance comparison with the DGCNN model in the subsequent experiments of graph convolutional layer analysis, which is the baseline of our HLGNN-MDA convolutional module.

The miRNA functional similarity and disease semantic similarity were downloaded directly from IMCMDA. HLGNN-MDA used 5430 known miRNA–disease associations as positive samples and a random sample of the same number of candidate associations as negative samples. The dataset with the positive and negative samples together was then randomly divided into ten parts. One copy of each round was selected as the test set, and the remaining nine copies were used as the training set. The tenfold cross-validation was completed in turn. In each round, HLGNN-MDA removed the positive samples in the test set from the adjacency matrix. All five comparison algorithms completed tenfold cross-validation on the miRNA–disease association matrix. There were six evaluation metrics to analyze the model: ACC, precision, recall, AUROC, AUPR and MCC. The results are shown in Table 1, and their corresponding ROC curves are shown in Figure 4. The corresponding precision–recall (PR) curves are shown in Figure 5.

In Table 1, HLGNN-MDA-hopx represents the HLGNN-MDA model in which the enclosing subgraph’s hop is x (x = 1, 2, 3 or 4). Hops were used to extract subgraphs. The extraction process is described in Section 3.2.1. Compared with other methods, HLGNN-MDA-hopx had high performance in five of the six indicators. The minimum AUROC of HLGNN-MDA-hopx was greater than those of BNPMDA, IMCMDA, LFEMDA and MKRMDA. The minimum value of the AUPR of HLGNN-MDA-hopx was greater than the results of all other algorithms.

From the ROC curve in Figure 4, HLGNN-MDA-hop4 could cover the curves of BNPMDA, IMCMDA and MKRMDA. The maximum AUROC was 0.25% larger than the AUROC of BLHARMDA. Similarly, in the PR curve of Figure 5, HLGNN-MDA-hop4 could cover the PR curves of BNPMDA, IMCMDA and MKRMDA, which were roughly the same as that of BLHARMDA. The maximum AUPR value of HLGNN-MDA-hop4 was 0.55% larger than the maximum value of BLHARMDA.

Compared with the other algorithms, HLGNN-MDA had a relatively large advantage. Moreover, HLGNN-MDA used less similarity information to obtain better prediction results.

### 2.2. Influence of Different Hops in the Enclosing Subgraph

In this section, we discuss how HLGNN-MDA was affected by different enclosing subgraph hops. The required experimental data were miRNA–disease associations. The positive samples represented the known associations, and a random sample of the same number of candidates represented the negative samples. The test set was one-tenth the size of the entire dataset.

First, we evaluated the results of the HLGNN-MDA model using different hops in the enclosed subgraph. Each model was trained with the same training set and evaluated on the same test set. The range of the number of hops in the enclosing subgraph was 1 to 4. Their corresponding results are shown in Table 2. The ROC, PR and accuracy curves under different thresholds are shown in Figure 6, Figure 7 and Figure 8.

As shown in Table 2, HLGNN-MDA obtained the best value for all six metrics when the hops were 4. In Figure 6, Figure 7 and Figure 8, it can be seen that the curves with less hops were always covered by curves with larger hops. In particular, it should be noted that the results of HLGNN-MDA-hop2 had a larger increase than the results of HLGNN-MDA-hop1. The results of HLGNN-MDA-hop4 had a larger increase than the results of HLGNN-MDA-hop3. There was only a slight increase between HLGNN-MDA-hop3 and HLGNN-MDA-hop2, as shown in the three figures. These findings show that using an even larger hop in enclosing subgraphs for miRNA–disease association prediction could obtain better results.

### 2.3. Analysis of the Improved Graph Convolutional Layer

To show that our improvement in the graph neural network was effective, in this section, we present the analysis of the improved graph convolutional layer. After considering the information transfer between homogeneous nodes and heterogeneous nodes, HLGNN-MDA aggregated four propagation functions in the graph convolutional layer: A, $A^{2}$, $D^{-1}A$ and $D^{-\frac{1}{2}}AD^{-\frac{1}{2}}$.

Next, we explored the role of each propagation function in graph neural networks. If we deleted a certain propagation function from the current HLGNN-MDA but obtained a better result, it showed that the propagation function had a negative effect on the graph neural network. If the result of deleting a certain propagation function was worse, it had a positive effect on the graph neural network. Therefore, we discussed the role of the four propagation functions in turn in this way. The HLGNN-MDA model that removed propagation function A was marked as HLGNN-MDA-a. In this order, the HLGNN-MDA model deleting $A^{2}$ was marked as HLGNN-MDA-b; the model without $D^{-1}A$ was marked as HLGNN-MDA-c; and the model without $D^{-\frac{1}{2}}AD^{-\frac{1}{2}}$ was marked as HLGNN-MDA-d.

Then, we compared the HLGNN-MDA model with its four variations: HLGNN-MDA-a, HLGNN-MDA-b, HLGNN-MDA-c and HLGNN-MDA-d. As shown in Figure 9, the horizontal axis is the hop in the enclosing subgraph, and the vertical axis is the AUROC. The red line indicates HLGNN-MDA, and the yellow line indicates HLGNN-MDA-b, wherein $A^{2}$ was deleted from HLGNN-MDA; these variations were the most disparate. These results showed that $A^{2}$ had a greater impact on the model. The difference between the green fold of HLGNN-MDA-a and the red fold of HLGNN-MDA was minimal. However, when the hop number was 1, 2 or 4, HLGNN-MDA gave better results than HLGNN-MDA-a. Meanwhile, HLGNN-MDA-a and HLGNN-MDA-b all reached a maximum with *hop* = 3 and decreased slightly with *hop* = 4. Both propagation functions A and $A^{2}$ played an effective role in improving the predictive performance of the HLGNN-MDA model. From the figures, we could find that $D^{-1}A$ (purple line) and $D^{-\frac{1}{2}}AD^{-\frac{1}{2}}$ (blue line) had similar roles in graph neural networks and complemented each other. Meanwhile, if the evaluation metrics in Figure 9 were replaced with the AUPR and the ACC, a similar trend could be obtained.

Furthermore, since the graph neural network [55,56] of HLGNN-MDA is improved with respect to DGCNN, in order to show the effectiveness of HLGNN-MDA model, we also compared the performance of four variants of HLGNN-MDA with DGCNN with different hops. From Table 3, it could be concluded that DGCNN and HLGNN-MDA-b performed similarly. HLGNN-MDA-b was slightly higher than DGCNN when the enclosing subgraph hops were 1, 2 and 3. Compared with other HLGNN-MDA variant models, HLGNN-MDA-b, i.e., the *A*^2^-deleted model, had the worst performance among all models. As the value of hops increased, the six measures of HLGNN-MDA-b also increased slowly, and when *hop* = 3, HLGNN-MDA-b reached a maximum. This finding was consistent with the results presented in the above figures; that is, $A^{2}$ played a large role in the HLGNN-MDA model. Overall, HLGNN-MDA performed better than DGCNN in predicting miRNA–disease associations.

In summary, the four propagation functions in HLGNN-MDA all played a positive role, thus leading HLGNN-MDA to achieve good predictive performance. Of these, A and $A^{2}$ had a stronger role in improving the predictive performance of the model, while $D^{-1}A$ and $D^{-\frac{1}{2}}AD^{-\frac{1}{2}}$ enabled the model to be stable in its results with different hops of the enclosing subgraph. The combined use of these four propagation functions allowed HLGNN-MDA to perform well in the task of predicting miRNA–disease associations.

### 2.4. Validation of Prediction Results

In this section, all the known correlations in HMDD v2.0 were the training set for the model, and as many potential associations as possible were sampled as negative samples. According to the above analysis, HLGNN-MDA-hop4 gave the best predictions, so training was performed on it. All potential association relationships in HMDD v2.0 were then extracted and predicted on the trained HLGNN-MDA-hop4.

The prediction results were validated using the following databases: HMDD v3.0 [57], dbDEMC [58] and miR2Disease [59]. One association was considered to be validated if it was found in at least one database.

The final validation results were as follows: 10 out of the top 10 predictions were validated; a total of 49 out of the top 50 predictions were verified; a total of 97 out of the top 100 predictions were verified; and 169 out of the top 180 predictions were verified. The results demonstrated the effectiveness of HLGNN-MDA and its ability to predict potential novel associations.

### 2.5. Case Study

#### 2.5.1. Breast Cancer

Breast cancer is one of the most dangerous malignancies to human health, especially for women. Globally, breast cancer accounts for 2.088 million new cases and 627,000 deaths per year, making it the number one malignancy in women [60]. The top 50 miRNAs predicted to be associated with breast cancer are listed in Table 4. In total, 49 of them were found in the relevant validation database. Only hsa-mir-362 (validation = no) does not present a record related to breast cancer in the three databases at present. Based on the literature validation [61], the hERG potassium channel, which enhances tumor aggressiveness and breast cancer proliferation, is transcriptionally regulated by hsa-miR-362-3p and thus associated with breast cancer growth. Another study [62] compared MDA-MB-231 and MCF7 breast cancer cell lines to the control CCD-1095Sk cell line, where hsa-miR-362-5p showed significant upregulation. The inhibition of hsa-miR-362-5p was found to significantly inhibit the diffusion, migration and invasion of MCF7 human breast cancer cells.

#### 2.5.2. Hepatocellular Carcinoma

Primary liver cancer is the fifth most common cancer worldwide, mainly including hepatocellular carcinoma (HCC) [63,64]. A total of 49 of the top 50 miRNAs predicted to be related to hepatocellular carcinoma could be validated in three validation databases. The results are shown in Table 5. At present, no clear association between hsa-mir-495 and hepatocellular carcinoma could be found in these databases. However, a previous study [65] reported that hsa-mir-495 expression was frequently downregulated in hepatocellular carcinoma tissues and cell lines. Its expression levels were significantly correlated with tumor size, tumor lymph node metastasis (TNM) stage and lymph node metastasis in patients with hepatocellular carcinoma [65].

#### 2.5.3. Renal Cell Carcinoma

Approximately 270,000 kidney cancer cases and 116,000 deaths are diagnosed annually worldwide [66]. Ninety percent of kidney cancers are tumors originating from the kidney epithelium and renal cell carcinoma [67]. The miRNAs predicted to be associated with the top 50 renal cell carcinomas are listed in Table 6. A total of 47 of the top 50 miRNAs could be validated clearly.

## 3. Materials and Methods

### 3.1. Data Resources

We collected human miRNA–disease associations from the HMDD v2.0 database [68] and obtained 5430 miRNA–disease associations between 495 miRNAs and 383 diseases. Therefore, many miRNA–disease associations were organized into an adjacency matrix *Y* ∈ *N*^*nm* × *nd*^, where nm and nd represent the number of miRNAs and the number of diseases, respectively. If an association between miRNA $m_{i}$ and disease $d_{j}$ was recorded in HMDD v2.0, then Y(i,j) equaled 1; otherwise, it equaled 0.

### 3.2. Methods

#### 3.2.1. Extraction of the Enclosing Subgraph of Node Pair

The design of our HLGNN-MDA model is inspired to the SEAL (learning from subgraphs, embedding and attributes for link prediction) [69] framework, which uses graph neural networks for link prediction. SEAL proves that most high-order heuristics can be approximated by learning from local enclosing subgraphs. Therefore, the first step of HLGNN-MDA is to extract the enclosing subgraph of all pairs of nodes. The enclosing subgraph is composed of two nodes, which are marked as central nodes, and their surrounding nodes. Then, the h-hop enclosing subgraph of central nodes m and d consists of all nodes whose distance from node m or node d is less than h steps.

HLGNN-MDA inputs the enclosing subgraphs into an “end-to-end” graph neural network for training. The association information of the central node pair of the enclosing subgraph is used as the supervisory label of the graph neural network. In association matrix Y, miRNA–disease associations equal to 1 are considered known, while those equal to 0 are treated as potential. The training set extracts known associations as positive samples and randomly samples an equal number of potential associations as negative samples. Subsequently, the enclosing subgraphs of all samples in the training set are extracted. To prevent the leakage of supervised labels, the edges between the central node pairs and the positive samples is removed during the extraction of the enclosing subgraph. The enclosing subgraph is denoted with A.

#### 3.2.2. Label Nodes

In this section, we label all the nodes in each enclosing subgraph. Node labeling is the process of assigning an integer to a node in an enclosing subgraph and can be defined as $f_{l}:V\to N$. Node labeling uses the DRNL (double-radius node labeling) method proposed in SEAL. First, we label the central two nodes with label 1. Other nodes with the same distance from the central two nodes are labeled with the same value. The farther the distance is, the greater the value is. The label of a node i can be derived from the following hash function:(1)$$f_{l}(i)=1+\min(d_{x},d_{y})+(d/2)[(d/2)+(d\%2)-1]$$
where $d_{x}$ and $d_{y}$ represent the distances from node i to central nodes x and y, respectively, with $d=d_{x}+d_{y}$; and d/2 and d%2 indicate division and remainder, respectively. When a node is disconnected from the central nodes, it is labeled with 0.

In this way, we can label all the nodes in the enclosing subgraphs. Then, before being input into the graph neural network, the label of each node is expanded into one-hot encoding as its feature. This one-hot feature represents the position information of the node in the enclosing subgraph.

#### 3.2.3. Construct Graph Neural Network

After extracting the enclosing subgraphs and labeling nodes in each enclosing subgraph, we can input the labeled enclosing subgraphs into the graph neural network for predicting miRNA–disease associations. At present, most of the proposed graph neural networks are applicable to homogeneous networks, whereas the miRNA–disease associations we use are in a bipartite graph network. Therefore, we improve the graph neural network DGCNN (deep graph convolutional neural network) to obtain better performance in heterogeneous networks.

**Graph convolution layers.** The role of the graph convolution layer is to learn the node representations [70,71]. A graph is input into the graph convolution layer through multilayer convolution, and the vector representation of each node can be extracted. The vector contains local substructural features of the graph. The process of graph convolution is the aggregation of feature information from the neighbors around each node.

Given the adjacency matrix of a graph $A\in R^{n\times n}$, its information matrix is $X\in R^{n\times d}$, where n indicates the number of nodes and d represents the dimension of the features. Graph convolution can be represented as follows:(2)Z=σ(f(A)⋅X⋅W)
where $W\in R^{d\times c}$ indicates the parameters to be trained, which converts the d-dimensional signal into c-dimensional signals; and f(A) represents the propagation function of adjacency matrix A. Usually, $f(A)={\tilde{D}}^{-1}\tilde{A}$ or $f(A)={\tilde{D}}^{-\frac{1}{2}}\tilde{A}{\tilde{D}}^{-\frac{1}{2}}$, where $\tilde{A}=A+I$ and $\tilde{D}$ are the degree matrices of $\tilde{A}$. Moreover, f(A)⋅X⋅W indicates that the feature vector of each node is aggregated in the manner of a propagation function and then undergoes information conversion. In addition, σ(⋅) is the activation function.

In the bipartite undirected network, miRNAs are only connected with diseases. That is, after a two-step jump, one disease node can only reach another disease, which is also true for miRNA nodes. Therefore, we use a propagation function for second-order topological information that allows information between homogeneous nodes to be aggregated together directly, i.e., defining a propagation function $f(A)=A^{2}$.

With different propagation functions selected, the topological characteristics of the graph learned by the graph convolutional network are slightly different [72]. By splicing the neighbor information of nodes aggregated by different propagation functions, the graph convolutional layer can learn better graph topological features. The graph convolutional layer of our HLGNN-MDA model is shown in Figure 3 and can be represented by Formula (3):(3)$$Z^{t+1}=\sigma \left(\left[A\cdot Z^{t},A^{2}\cdot Z^{t},D^{-1}A\cdot Z^{t},D^{-\frac{1}{2}}AD^{-\frac{1}{2}}\cdot Z^{t}\right]\cdot W^{t}\right)$$
where $Z^{0}=X$ and $Z^{t}\in R^{n\times d_{t}}$ represent the output of the t-th graph convolutional layer; $d_{t}$ is the output dimension of the t-th layer graph convolution; and [⋅] represents the splicing of row vectors, which splices the node vectors obtained through different propagation functions. The propagation functions used in our model are A, $A^{2}$, $D^{-1}A$ and $D^{-\frac{1}{2}}AD^{-\frac{1}{2}}$. After splicing, the topological features captured by these several propagation functions can be considered at the same time. $W^{4d_{t}\times d_{t+1}}$ is the parameter to be trained, which maps the spliced node features from the $4d_{t}$ dimension to the $d_{t+1}$ dimension. Dimension $4d_{t}$ is used here because there are four propagation functions involved in the graph convolution process.

Enclosing subgraph A and its node features X go through t graph convolution layers to produce output $Z^{t}$, t=1,…,T. The overall graph convolution layer uses a global structure where the results of each layer are stitched together at the end, resulting in a result denoted as $Z=[Z^{1},\ldots ,Z^{T}]$. Each row of $Z\in R^{n\times \sum_{1}^{T}d_{t}}$ is the $\sum_{1}^{T}d_{t}$-dimensional embedding vector representation of a node that contains rich topological information about that node in that graph. Figure 2 shows the overall architecture of the graph convolutional layer.

**Graph pooling layers.** A graph convolutional layer is used to learn a latent vector representation for each node. Here, the graph pooling layer of HLGNN-MDA selects the k most important nodes from the nodes of the graph to represent the graph. The importance of the node is evaluated using the result of the graph convolutional layer. The last layer of graph convolution maps the result of the previous layer to 1 dimension. That is, the parameter, $W^{T}$, of the last layer of graph convolution maps dimension $4d_{T-1}$ to 1 dimension. In this way, a value is obtained for each node, and this value indicates how important the node is in the graph.

We sort Z in descending order according to its last dimension. If two nodes appear to be equal in the last dimension of Z, we compare their penultimate dimension and so on, until the two nodes can be separated. The graph pooling layer takes the top k nodes in the ranking as its output, which helps subsequent conventional neural network layers to obtain a tensor with a fixed specification. When the number of nodes, n, is less than k, (n−k) zero vectors are added after the sorted nodes.

**Convolution layers and fully connected layer.** After the graph pooling layer, a tensor $Z=k\times \sum_{1}^{T}d_{t}$ is obtained. In the remaining layers, we first use the traditional one-dimensional convolutional neural network combined with the max pooling layer to further refine the graph representation features and then make the final prediction using a fully connected layer.

To train a one-dimensional convolutional neural network on tensor Z, it first needs to be reshaped into a one-dimensional vector. To apply the filter for each node feature, the filter size and step length of the first one-dimensional convolutional neural network are set to $\sum_{1}^{T}d_{t}$; that is, each node feature is convolved first. Then, after a max pooling layer, a one-dimensional convolutional neural layer is used to further learn the graph to represent the local features in the sequence features. Finally, it is connected to the fully connected layer. We use NLLLoss as the loss function, which adds up the predicted values of all predicted samples under the true labels. The predicted value here is a negative number in the logarithmic form of a normalized exponential function (softmax), mapping the prediction range from (0,1) to (0,+∞). If all prediction samples are predicted correctly, NLLLoss is closer to 0. Finally, the fully connected layer outputs the probability of miRNA and disease node pair connections.

#### 3.2.4. Evaluation Metrics

In order to verify the model performance, we choose the following metrics: AUROC (Receiver Operating Characteristic curve), ACC (accuracy), precision, recall, AUPR (area under the precision–recall curve) and MCC (Matthews correlation coefficient). The relevant definitions are as follows:(4)$$\mathrm{Precision}=\frac{\mathrm{TP}}{\mathrm{TP}+\mathrm{FP}}$$
(5)$$\mathrm{Recall}=\frac{\mathrm{TP}}{\mathrm{TP}+\mathrm{TN}}$$
(6)$$\mathrm{FPR}=\frac{\mathrm{FP}}{\mathrm{FP}+\mathrm{TN}}$$
(7)$$\mathrm{ACC}=\frac{\mathrm{TP}+\mathrm{TN}}{\mathrm{TP}+\mathrm{TN}+\mathrm{FP}+\mathrm{FN}}$$
(8)$$\mathrm{MCC}=\frac{\mathrm{TP}-\mathrm{TN}\times \mathrm{FP}\times \mathrm{FN}}{\sqrt{\left(\mathrm{TP}+\mathrm{FP})(\mathrm{TP}+\mathrm{FN})(\mathrm{TN}+\mathrm{FP})(\mathrm{TN}+\mathrm{FN}\right)}}$$

TP (true positive), FP (false positive), TN (true negative) and FN (false negative) were all derived from the confusion matrix. The ROC curve has the FPR (false positive rate) and TPR (recall or true positive rate) as the horizontal and vertical coordinates, respectively, and the area under the ROC curve is the AUC value. The area under the curve with recall and precision as the horizontal and vertical coordinates is the AUPR value.

## 4. Conclusions

Understanding the relationship between miRNAs and disease has important implications for disease prevention, detection and treatment. This paper proposes the HLGNN-MDA method, which is a heuristic for learning miRNA–disease association prediction from known miRNA–disease relationships based on graph neural networks. HLGNN-MDA first extracts the enclosing subgraphs around each miRNA–disease pair to be predicted to obtain the local network structure. Each node in the enclosing subgraphs is then labeled. The labeled subgraphs are then input into the graph neural network for classification. In particular, second-order topological information is added to the convolutional layer of the graph neural network to enable it to learn information between similar nodes. Second, different combinations of propagation functions are designed to improve the accuracy and stability of the graph neural network. We compared the model with the same type of miRNA–disease association prediction model using tenfold cross-validation. The results showed that HLGNN-MDA was able to obtain better performance than most miRNA–disease association prediction models. After discussing the effect of hop count on extracting closed subgraphs, we successively evaluated each propagation function combined in the model. Finally, we used the trained HLGNN-MDA model to make predictions and performed case studies on breast cancer, hepatocellular carcinoma and renal cell carcinoma. A total of 49 of the top 50 predicted miRNAs for breast cancer could be found in the validation database. The remaining hsa-mir-362 was also found to be associated with breast cancer, as supported by the literature. Similarly, 49 of the top 50 miRNAs predicted for hepatocellular carcinoma could be validated against the database. The remaining hsa-mir-495 was also found to be related to hepatocellular carcinoma, as supported by the literature. Finally, 47 of the top 50 miRNAs associated with renal cell carcinoma could be validated.

Generally, HLGNN-MDA has the following advantages: First, HLGNN-MDA can select arbitrary links for prediction without having to predict all potential miRNA–disease associations in the adjacency matrix. In particular, when predicting individual diseases or miRNAs, HLGNN-MDA can directly obtain the corresponding results. Second, HLGNN-MDA does not strictly require the corresponding similarity data because it can learn information through the topology of the network.

However, there are still some aspects for improvement. Valid miRNA and disease signatures are also important for prediction. Therefore, adding valid miRNA and disease signatures to HLGNN-MDA should be further investigated. Second, HLGNN-MDA is an end-to-end supervised learning algorithm framework. One of the problems with miRNA–disease association prediction is that there is no definite negative sample. For the potential associations, only a small proportion of them are truly associated, and most are unassociated. In this paper, the prediction of miRNA–disease association is approximated as a supervised learning model [73] with insufficient samples. Therefore, a new linkage prediction heuristic represents a future researchable direction.

## Figures and Tables

**Figure 2 ijms-23-13155-f002:**
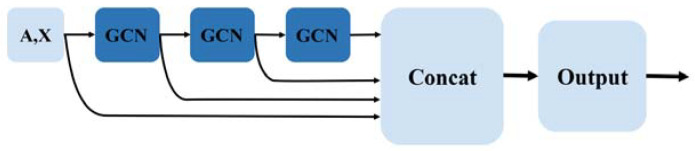
Three-layer graph convolutional layer structure of HLGNN-MDA.

**Figure 3 ijms-23-13155-f003:**
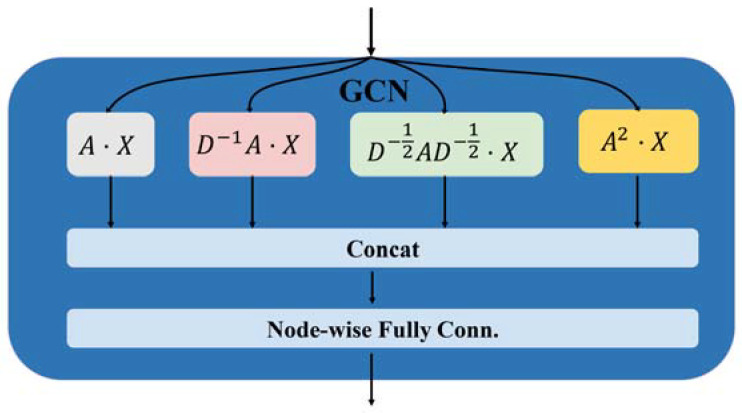
Design of the graph convolution block for HLGNN-MDA.

**Figure 4 ijms-23-13155-f004:**
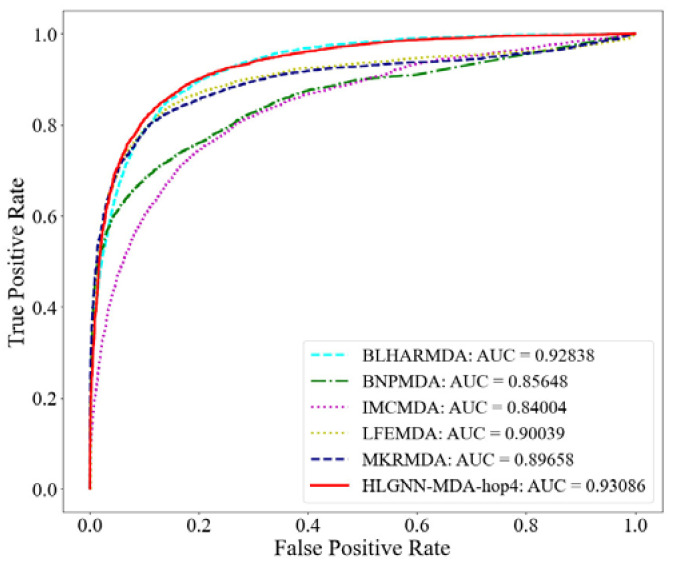
ROC curve of HLGNN-MDA and state-of-the-art miRNA–disease association prediction algorithms.

**Figure 5 ijms-23-13155-f005:**
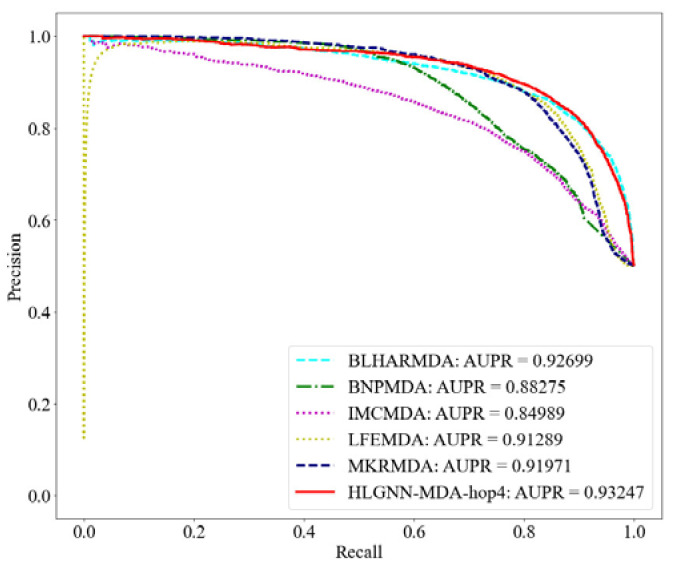
PR curve of HLGNN-MDA and state-of-the-art miRNA–disease association prediction algorithms.

**Figure 6 ijms-23-13155-f006:**
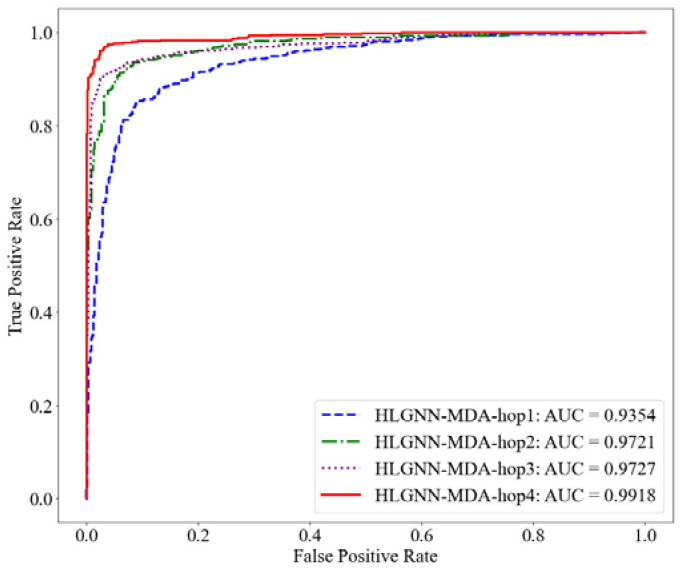
ROC curve of HLGNN-MDA in different enclosing subgraphs.

**Figure 7 ijms-23-13155-f007:**
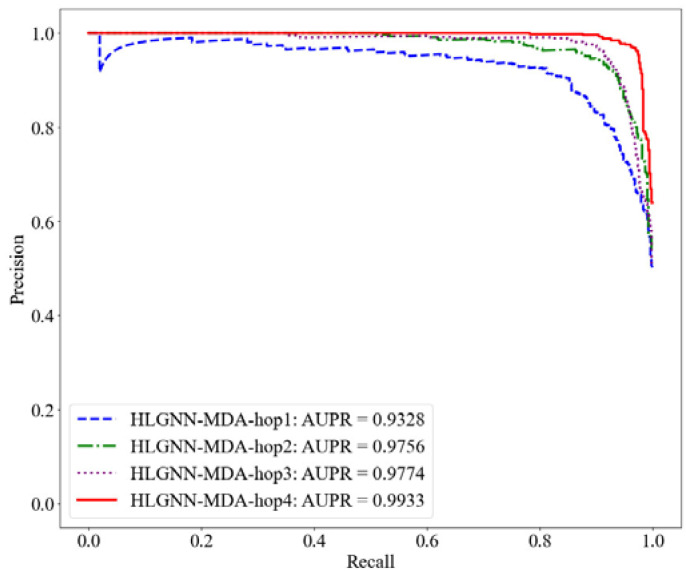
PR curve of HLGNN-MDA in different enclosing subgraphs.

**Figure 8 ijms-23-13155-f008:**
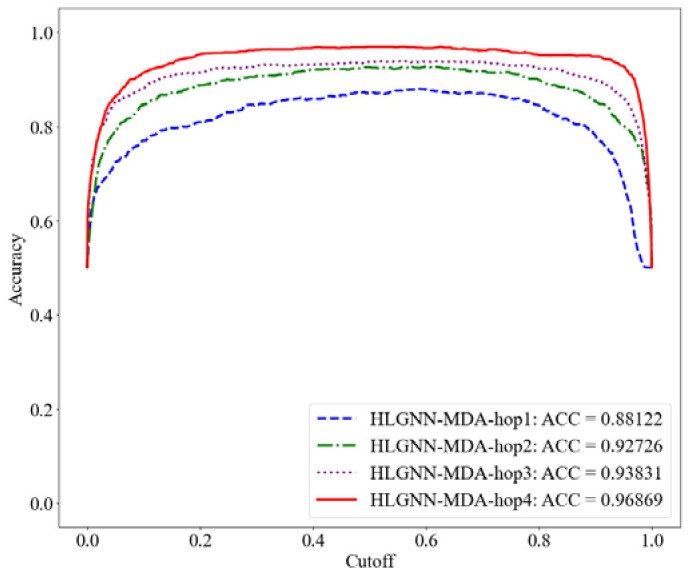
Accuracy curve of HLGNN-MDA using different enclosing subgraph hops.

**Figure 9 ijms-23-13155-f009:**
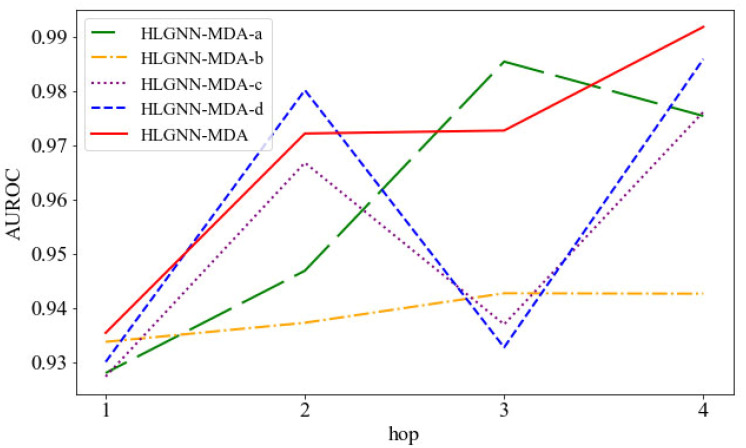
AUC comparison between HLGNN-MDA and its four variations under different hops.

**Table 1 ijms-23-13155-t001:** Performance analysis of HLGNN-MDA and state-of-the-art miRNA–disease association prediction algorithms. The bold part indicates the maximum value in the corresponding column.

Model	ACC	Precision	Recall	AUROC	AUPR	MCC
BNPMDA	0.79088	0.87069	0.68324	0.85648	0.88275	0.59574
IMCMDA	0.77274	0.80102	0.72578	0.84004	0.84989	0.54791
LFEMDA	0.84751	0.85590	0.83573	0.90039	0.91289	0.69522
BLHARMDA	0.85442	0.85619	0.85193	0.92838	0.92699	0.70885
MKRMDA	0.84549	**0.87610**	0.80479	0.89658	0.91971	0.69328
HLGNN-MDA-hop1	0.85442	0.86263	0.84309	0.92974	0.92779	0.70902
HLGNN-MDA-hop2	**0.85976**	0.85917	**0.86059**	0.92833	0.92927	**0.71952**
HLGNN-MDA-hop3	0.85635	0.86745	0.84125	0.92863	0.93007	0.71303
HLGNN-MDA-hop4	0.85912	0.86709	0.84825	**0.93086**	**0.93247**	0.71840

**Table 2 ijms-23-13155-t002:** Influence of different hops in enclosing subgraphs on HLGNN-MDA. The bold part indicates the maximum value in the corresponding column.

Model	ACC	Precision	Recall	AUROC	AUPR	MCC
HLGNN-MDA-hop1	0.88122	0.90430	0.85267	0.93535	0.93281	0.76368
HLGNN-MDA-hop2	0.92726	0.93939	0.91344	0.97212	0.97564	0.85484
HLGNN-MDA-hop3	0.93831	0.95946	0.91529	0.97266	0.97744	0.87754
HLGNN-MDA-hop4	**0.96869**	**0.96690**	**0.97053**	**0.99178**	**0.99332**	**0.93739**

**Table 3 ijms-23-13155-t003:** Comparison between the four variations of HLGNN-MDA and DGCNN with different hops.

Model	ACC	Precision	Recall	AUROC	AUPR	MCC
HLGNN-MDA-a-hop1	0.85820	0.91121	0.79374	0.92795	0.92303	0.72242
HLGNN-MDA-a-hop2	0.90055	0.90503	0.89503	0.94681	0.94629	0.80115
HLGNN-MDA-a-hop3	0.94843	0.94516	0.95212	0.98538	0.98626	0.89689
HLGNN-MDA-a-hop4	0.93186	0.95183	0.90976	0.97535	0.97945	0.86456
HLGNN-MDA-b-hop1	0.86096	0.90164	0.81031	0.93369	0.93412	0.72565
HLGNN-MDA-b-hop2	0.87845	0.92371	0.82505	0.93721	0.94106	0.76126
HLGNN-MDA-b-hop3	0.89042	0.91569	0.86004	0.94265	0.94575	0.78229
HLGNN-MDA-b-hop4	0.88858	0.90734	0.86556	0.94257	0.94877	0.77799
HLGNN-MDA-c-hop1	0.85635	0.84127	0.87845	0.92729	0.92241	0.71340
HLGNN-MDA-c-hop2	0.92265	0.91652	0.93002	0.96673	0.96729	0.84540
HLGNN-MDA-c-hop3	0.88398	0.92464	0.83610	0.93691	0.94300	0.77150
HLGNN-MDA-c-hop4	0.93923	0.96311	0.91344	0.97610	0.97901	0.87962
HLGNN-MDA-d-hop1	0.86280	0.89879	0.81768	0.92999	0.92586	0.72857
HLGNN-MDA-d-hop2	0.93831	0.94238	0.93370	0.98012	0.98081	0.87665
HLGNN-MDA-d-hop3	0.87569	0.92324	0.81952	0.93266	0.94060	0.75617
HLGNN-MDA-d-hop4	0.94015	0.95437	0.92449	0.98585	0.98702	0.88073
DGCNN-hop1	0.85820	0.88822	0.81952	0.92889	0.92889	0.71854
DGCNN-hop2	0.87201	0.90400	0.83241	0.93509	0.93831	0.74636
DGCNN-hop3	0.88582	0.90522	0.86188	0.94241	0.94083	0.88302
DGCNN-hop4	0.89411	0.91961	0.86372	0.95250	0.95707	0.89079

**Table 4 ijms-23-13155-t004:** Top 50 miRNAs predicted by HLGNN-MDA to be associated with breast cancer.

Rank	MicroRNA	Validation	Rank	MicroRNA	Validation
1	hsa-mir-211	yes <H, D>	26	hsa-mir-30e	yes <H, D>
2	hsa-mir-186	yes <D>	27	hsa-mir-494	yes <H, D>
3	hsa-mir-744	yes <H, D>	28	hsa-mir-421	yes <H, D>
4	hsa-mir-138	yes <H, D>	29	hsa-mir-501	yes <H, D>
5	hsa-mir-154	yes <D>	30	hsa-mir-99b	yes <H, D>
6	hsa-mir-216b	yes <H, D>	31	hsa-mir-196b	yes <H, D>
7	hsa-mir-106a	yes <H, D>	32	hsa-mir-185	yes <H, D>
8	hsa-mir-432	yes <H, D>	33	hsa-mir-484	yes <H, D>
9	hsa-mir-32	yes <H, D>	34	hsa-mir-144	yes <H, D>
10	hsa-mir-381	yes <H, D>	35	hsa-mir-592	yes <H, D>
11	hsa-mir-142	yes <H, D>	36	hsa-mir-130a	yes <H, D>
12	hsa-mir-150	yes <H, D>	37	hsa-mir-542	yes <H, D>
13	hsa-mir-491	yes <H, D>	38	hsa-mir-1224	yes <H, D>
14	hsa-mir-449a	yes <H, D>	39	hsa-mir-376a	yes <H, D>
15	hsa-mir-362	no	40	hsa-mir-451	yes <H, D, M>
16	hsa-mir-28	yes <H, D>	41	hsa-mir-433	yes <H, D>
17	hsa-mir-378a	yes <H, D>	42	hsa-mir-483	yes <H, D>
18	hsa-mir-212	yes <H, D>	43	hsa-mir-1207	yes <H, D>
19	hsa-mir-98	yes <H, D, M>	44	hsa-mir-33b	yes <H, D>
20	hsa-mir-92b	yes <H, D>	45	hsa-mir-15b	yes <H, D>
21	hsa-mir-455	yes <H, D>	46	hsa-mir-630	yes <H, D>
22	hsa-mir-590	yes <H, D>	47	hsa-mir-622	yes <H, D>
23	hsa-mir-330	yes <H, D>	48	hsa-mir-1271	yes <H, D>
24	hsa-mir-675	yes <H, D>	49	hsa-mir-424	yes <H, D>
25	hsa-mir-217	yes <H, D>	50	hsa-mir-95	yes <H, D>

Note: H <HMDD v3.0>, D <dbDEMC> and M <miR2Disease > represent the databases in which the relations could be validated.

**Table 5 ijms-23-13155-t005:** Top 50 miRNAs predicted by HLGNN-MDA to be associated with hepatocellular carcinoma.

Rank	MicroRNA	Validation	Rank	MicroRNA	Validation
1	hsa-mir-143	yes <H, D, M>	26	hsa-mir-23b	yes <H, D, M>
2	hsa-mir-196b	yes <H, D>	27	hsa-mir-574	yes <H, D>
3	hsa-mir-137	yes <H, D, M>	28	hsa-mir-26b	yes <H, D, M>
4	hsa-mir-520c	yes <H, D>	29	hsa-mir-495	no
5	hsa-mir-376c	yes <H, D>	30	hsa-mir-328	yes <H, D, M>
6	hsa-mir-184	yes <H, D>	31	hsa-mir-452	yes <H, D>
7	hsa-mir-215	yes <H, D, M>	32	hsa-mir-204	yes <H, D>
8	hsa-mir-302a	yes <H, D>	33	hsa-mir-135b	yes <H, D>
9	hsa-mir-34b	yes <H, D>	34	hsa-mir-95	yes <H, D>
10	hsa-mir-339	yes <H, D>	35	hsa-mir-185	yes <H, D, M>
11	hsa-mir-708	yes <H, D>	36	hsa-mir-206	yes <H, D>
12	hsa-mir-193	yes <H, D>	37	hsa-mir-449a	yes <H, D>
13	hsa-mir-30e	yes <H, D, M>	38	hsa-mir-520a	yes <H, D>
14	hsa-mir-488	yes <H, D>	39	hsa-mir-194	yes <H, D, M>
15	hsa-mir-200	yes <H, M>	40	hsa-mir-451	yes <H, D>
16	hsa-mir-342	yes <H, D>	41	hsa-mir-149	yes <H, D>
17	hsa-mir-367	yes <H, D>	42	hsa-mir-153	yes <H, D>
18	hsa-mir-302d	yes <H, D>	43	hsa-mir-299	yes <H, D>
19	hsa-mir-494	yes <H, D>	44	hsa-mir133a	yes <H, D, M>
20	hsa-mir-128	yes <H, D, M>	45	hsa-mir-633	yes <D>
21	hsa-mir-340	yes <H, D>	46	hsa-mir-132	yes <H, D, M>
22	hsa-mir-33b	yes <H, D>	47	hsa-mir-27b	yes <H, D>
23	hsa-mir-625	yes <H, D>	48	hsa-mir-935	yes <H, D>
24	hsa-mir-424	yes <H, D>	49	hsa-mir-32	yes <H, D>
25	hsa-mir-151b	yes <H, D>	50	hsa-mir-186	yes <H, D, M>

Note: H <HMDD v3.0>, D <dbDEMC> and M <miR2Disease > represent the databases in which the relations could be validated.

**Table 6 ijms-23-13155-t006:** Top 50 miRNAs predicted by HLGNN-MDA to be associated with renal cell carcinoma.

Rank	MicroRNA	Validation	Rank	MicroRNA	Validation
1	hsa-mir-20a	yes <H, D, M>	26	hsa-mir-181a	yes <H, D>
2	hsa-mir-17	yes <H, D, M>	27	hsa-mir-192	yes <H, D>
3	hsa-mir-27b	yes <H, D>	28	hsa-mir-22	yes <H, D>
4	hsa-mir-221	yes <H, D, M>	29	hsa-mir-182	yes <H, D, M>
5	hsa-mir-223	yes <H, D, M>	30	hsa-mir-29b	yes <H, D, M>
6	hsa-mir-31	yes <H, D>	31	hsa-mir-15a	yes <H, D, M>
7	hsa-mir-29a	yes <H, D, M>	32	hsa-mir-375	yes <H, D>
8	hsa-mir-125b	yes <H, D>	33	hsa-mir-486	yes <D>
9	hsa-mir-133a	yes <H, D, M>	34	hsa-mir-15b	yes <H, D>
10	hsa-mir-125a	yes <H, D>	35	hsa-mir-107	yes <H, D>
11	hsa-mir-18a	yes <H, D>	36	hsa-mir-328	yes <D>
12	hsa-mir-1	yes <H, D>	37	hsa-mir-23a	yes <D>
13	hsa-mir-30a	yes <H, D, M>	38	hsa-mir-194	yes <H, D>
14	hsa-mir-181b	yes <H, D>	39	hsa-mir-193b	yes <H, D>
15	hsa-mir-19b	yes <H, D, M>	40	hsa-mir-196b	yes <D>
16	hsa-mir-214	yes <H, D, M>	41	hsa-mir-137	yes <H, D>
17	hsa-mir-130a	yes <H, D>	42	hsa-mir-191	yes <H, D, M>
18	hsa-mir-222	yes <H, D>	43	hsa-mir-302a	no
19	hsa-mir-148a	yes <H, D>	44	hsa-mir-135b	yes <D>
20	hsa-mir-25	yes <D>	45	hsa-mir-451b	no
21	hsa-mir-133b	yes <H, D>	46	hsa-mir-342	yes <D, M>
22	hsa-mir-183	yes <H, D>	47	hsa-mir-30b	yes <H, D>
23	hsa-mir-106a	yes <H, D, M>	48	hsa-mir-373	no
24	hsa-mir-24	yes <D>	49	hsa-mir-212	yes <D>
25	hsa-mir-132	yes <D>	50	hsa-mir-193a	yes <H, D>

Note: H <HMDD v3.0>, D <dbDEMC> and M <miR2Disease > represent the databases in which the relations could be validated.

## Data Availability

Code of the model, datasets and results can be downloaded from GitHub (https://github.com/LiangYu-Xidian/HLGNN-MDA).

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
