# Peer review of "HLGNN-MDA: Heuristic Learning Based on Graph Neural Networks for miRNA–Disease Association Prediction"

_ijms, 2022, doi:10.3390/ijms232113155_

Round 1
Reviewer 1 Report
Recommendation: Author should prepare a major revision
In this paper, the authors presented a disease-related miRNA prediction method called HLGNN-MDA.
HLGNN-MDA applied graph neural networks by learning local graph topology feature of the predicted miRNA-disease node.
1. The authors should revise English writing carefully and eliminate small typos ( e.g. dis-ease, bio-logical, per-forming) in the paper to make the paper easier to understand.
2. Recently, HMDD released latest version of HMDD v3.2. It is recommened to provide experimental results based on HMDD v.3.2.
3. Literature review is incomplete. The authors should provide reviews for previous miRNA-disease association prediction computational models published in the journals, such as the paper with PMIDs: 30142158, 35743670 and DOI: 10.1109/ACCESS.2021.3084148.
4. It is suggested to carry out another comparitive experiment. Please compare HLGNN-MDA with other preivious methods (PMID: 34020550, 31857202) to predict the MDAs in the framework of global and local LOOCV.
Author Response
请查看附件
Reviewer 2 Report
In this manuscript, the authors aim to accurately predict MDA by improving graph neural network through a heuristic learning approach using the enclosing subgraphs. The method is reasonable; however, multiple concerns including a few major concerns are listed below (1, 4, 11, 12, 14, 16), especially the 14th comment:
1. English needs significant improvement. It is hard to read.
2. The introduction misses the part about GNN and its application on MDA. Such as MDA-GCNFTG. Some information was presented in the method section, which can be re-organized.
3. At Line 71: It is not proper to say “the existing semantic similarity of diseases is of little use for prediction” since it was used in many previous MDA methods as disease features.
4. Why were the five methods selected for comparison? The author should include the previous developed GCN-based methods.
5. Line 279: The HMDD v3 is available a few years ago (2018), not sure why the authors use the data from v2.0.
6. The legend for the Figure 10 is too simple. In Figure 10b “Graph convolution layers”, what are the differences between the three GCN subgraphs before merging into one? Can the author clarify why the figure 11 embedded in figure 10b only point to one of the three subgraphs? What do the colors and the labels in Figure 10b means, such as A, C, D with different colors? Were the labels changed from input to GCN?
7. The legend for Figure 12 is too simple. Can the author clarify the differences of the three GCN layers? Is the GCN in Figure 11 one of the three GCNs in Figure 12? Why is the input directly concatenated with the three GCN layers in Figure 12?
8. In the three Case Studies, can the author list the validated dataset name in the results tables? If most of the results are only validated by the HMDD v3.2 dataset, it will be not convincing.
9. Since there were no definite negative samples, can the author perform multiple times of random sampling and examine how they impact the results?
10. What is the performance if the hop is 5 and 6? Increasing the number of hops essentially increase the size of subgraphs. How does the cost on the running time? What is the performance and running for taking the whole graph as input?
11. The test on the number of hops should present before the comparison with other methods. The author did not state what hop is and why multiple hops were used at the first section of the results but directly present the results in table 1. It is not fair for other methods if use the optimized parameter for HLGNN-MDA but not for other methods.
12. What are the data used for the hop number examination and the comparison with other methods? The results for different number of hops were different (Table 1 vs Table 2; so did for the related figures). Are the input data the same for table 1 and table 2? Or it is caused by sampling process?
13. Can the author explain why even hop gives better results? Does it mean that HLGN-MDA is better to capture homogenous information?
14. What are the differences between results sections 2.2 and 2.3, why does the author repeat a whole section and duplicated many figures?
15. DGCNN should be included at the first section of the results for comparison too.
16. In results Section 2.5, what is the result after removing HMDD v3.0? The dataset from the same source should not be used as an effective independent validation dataset.
Round 2
Reviewer 1 Report
Accept the present form of paper without any additional modifications
Author Response
Thank you!
Reviewer 2 Report
The authors did not response to my questions carefully for most of my questions:
1, The English did not change much. I cannot see major language changes.
2, The author simply included the MDA-GCNFTG, but did not say anything about the differences of the GCN method used in MDA-GCNFTG with their GCN method.
4, The authors did not compare their method with any existing GCN method.
6, The authors did not add any further legends. And they did not modify their figure to make it more clear to the readers.
7, The author did not explain why the input is directly connected with other GCN layers.
8, The author did not add the validated datasets into the tables, which is a very easy work to do. I don't understand why the authors reject to do it.
9, The author did not add anything about the negative sampling in the revised manuscript although they simply said in replying my question that they did. I can not see where they presented the results or stated it in their revised manuscript.
10, The explanation about the impact of hops is not clear. According to the current conclusion, the highest hop the authors used gave the best results (hop 4) and they rejected to test higher hops. Therefore, the subgraphs may not work well compared to using the whole graph.
14, The author did not present “Analysis of the improved graph convolutional layer” in the first version of the manuscript. They should acknowledge this first.
15, The author is not familiar with their work. They did the comparison with DGCNN in their manuscript but claimed "not" here in replying my question. I did not see which method is the same type of DGCNN as they claimed here. For this question, I simply asked the author to include DGCNN in their first section of results, which is very simple to do since they compared with DGCNN in the later part of the manuscript. I can not understand why the authors refuse to do it by claiming they did not even compared with DGCNN. They have compared with DGCNN at the results section already.
16, The author did not remove HMDD v3.0 for their case study and see how it impacts their results.
